# Intra- and Inter-Annual Variability in the Dissolved Inorganic Nitrogen in an Urbanized River before and after Wastewater Treatment Plant Upgrades: Case Study in the Grand River (Southwestern Ontario)

**Eduardo Cejudo** [1,2,*] ⦾**, Madeline S. Rosamond** [1]**, Richard J. Elgood** [1] **and Sherry L. Schiff** [1]

[1] Department of Earth Sciences, University of Waterloo, 200 University Ave. West, Waterloo, ON N2L 3G1, Canada; msrosamo@uwaterloo.ca (M.S.R.); rjelgood@uwaterloo.ca (R.J.E.); sschiff@uwaterloo.ca (S.L.S.)
[2] CONACYT–Centro de Investigación Científica de Yucatán A.C., Unidad de Ciencias del Agua, 97205 Merida, Mexico
* Correspondence: eduardo.cejudo@cicy.mx; Tel.: +52-998-2113008 (ext. 121)

**Abstract:** External nitrogen (N) inputs originating from human activities act as essential nutrients accumulation in aquatic ecosystems or it is exported elsewhere, where the assimilation capacity is surpassed. This research presents a multi-annual case study of the dissolved inorganic nitrogen (DIN) in an urban river in Ontario (Canada), assessed changes in N downstream of the largest wastewater treatment plant (WTP) in the watershed. Changes in the DIN effluent discharge, in-river concentrations and loads were observed comparing the intra- and inter-annual variability (2010–2013) before, during and after WTP upgrades. These upgrades reduced the ammonium concentration in the river from 0.44 to 0.11 mg $N-NH_4^+$/L (year average), but the N load in the effluent increased. In the river, nitrate and ammonium concentrations responded to seasonal variability, being higher during the low temperature (>10 °C) and high flow seasons (spring and spring melt). Among years, changes in the DIN concentration are likely controlled by the effluent to river dilution ratio, which variability resides on the differences in river discharge between years. This suggest that the increasing trend in the DIN concentration and loads are the result of agricultural and urban additions, together with reduced N assimilation, in addition to N loads responding to variable river discharge. Finally, we propose monitoring both concentrations and loads, as they provide answers to different questions for regulatory agencies and water managers, allowing tailored strategies for different purposes, objectives and users.

**Keywords:** dissolved inorganic nitrogen; effluent; seasonality; urban; wastewater

## 1. Introduction

Nitrogen (N) is an essential nutrient that is sometimes limiting in aquatic ecosystems [1–4]. In most aquatic ecosystems, primary producers meet their N requirements by a combination of N fixation, mineralization, atmospheric deposition and surface runoff [5]. Additional inputs of N can promote increasing primary productivity, greenhouse gas production, and N accumulation or export, which all raise ecological, managerial and regulatory concerns [6]. The global average N concentration in rivers ranges from 0.12 mg N/L in ecosystems with little human influence, to more than 1 mg N/L in rivers with high human activity [7,8]. Most of the dissolved inorganic nitrogen (DIN = $NO_2^-$ + $NO_3^-$ + $NH_4^+$) entering into rivers and streams originates from human activities [9–11]. Surplus N inputs can be from point sources (wastewater effluent, septic systems) and non-point sources (agricultural runoff, atmospheric deposition). Thus, the capacity of a catchment to process, retain or export reactive nitrogen depends primarily on its in-stream productivity, nutrient availability, channel morphology and river discharge [12–14].

Urban rivers commonly have high N, namely from domestic wastewater [15]; however, some rivers also receive agricultural inputs, such as the Grand River [16]. N concentration is used as a descriptor of the overall ecosystem health; it is relevant for regulatory agencies dealing with water quality guidelines, toxicity, environmental compliance limits and ecosystem health [17]. In addition, monitoring N concentrations tracks the changes in water quality over time due to land use or anthropogenic impacts and collects information useful in planning nutrient management in agricultural and urban watersheds [18,19]. However, flux estimates are frequently used for assessing N budgets, export and subsidies, productivity gradients and evaluating the effectiveness management practices and policies [20,21].

The Grand River (southwestern Ontario, Canada) receives N inputs from 30 municipal wastewater treatment plants (WTPs) and intensive agriculture [22]. An early characterization of the watershed [23] identified WTPs as sources of nutrients and potential threats to the trophic status of the river. Recent studies in the Grand River regarding nutrients cycling [24–29] showed that N inputs from point sources are of great concern, needing special attention. In order to improve the quality of the wastewater effluents and maintain the ecosystem integrity of the Central Grand River (an urbanized area with relatively high population and several WTPs), the Region of Waterloo implemented upgrades in the two largest treatment plants by volume in the Grand River watershed. The Kitchener Wastewater Treatment Plant (KWTP) is a conventional activated sludge plant comprised of two separate secondary treatment plants with average day capacity of 122,745 m$^3$/d. The upgrades comprised nitrified effluent (submerged aeration), biosolids dewatering, UV disinfection, achieving full nitrification during 2013 [28,30]. The Waterloo Wastewater Treatment Plant (WWTP) is also a conventional activated sludge plant with average-day capacity of 57,500 m$^3$/d. This wastewater treatment plant underwent a less extent upgrading including increased oxygenation, digester upgrades, thickening and dewatering [31].

The objective of this paper is to present an interpretation of a multi-annual study of the DIN dynamics in an urbanized river, exemplified by the central Grand River, Ontario. Our approach accounted for changes in the quality of WTP effluent discharging into the central Grand River, considering seasonal variability (intra annual) and a comparison before, during and after upgrades (inter annual) of the largest WTP by volume in the watershed.

## 2. Materials and Methods

### 2.1. Site Description

The Grand River watershed (280 km long, 6800 km$^2$) is divided into three sections: (i) northern till plains, (ii) central moraines and (iii) southern plains with lacustrine influence [32]. The river flows approximately south into Lake Erie. This research focused on the central region, where land use is both agricultural and urban. Biweekly or three-week periodical sampling was completed from 2010 to 2013 at four locations in the central Grand River, from West Montrose in the north (43.5856 N, −80.4816 W, 98 km from headwaters) to Brantford in the south (43.1523 N, −80.3173 W, 204 km from headwaters; Figure 1). The sampling locations representing upstream conditions (relative to the urban area) were West Montrose (WM) and Bridgeport (BR). The location representing the urban area downstream of the Kitchener Wastewater Treatment Plant was Blair (BL), located 5.7 km downstream of the effluent discharge. Finally, Brant Conservation Area (BCA) was the location representing the site with cumulative effects.

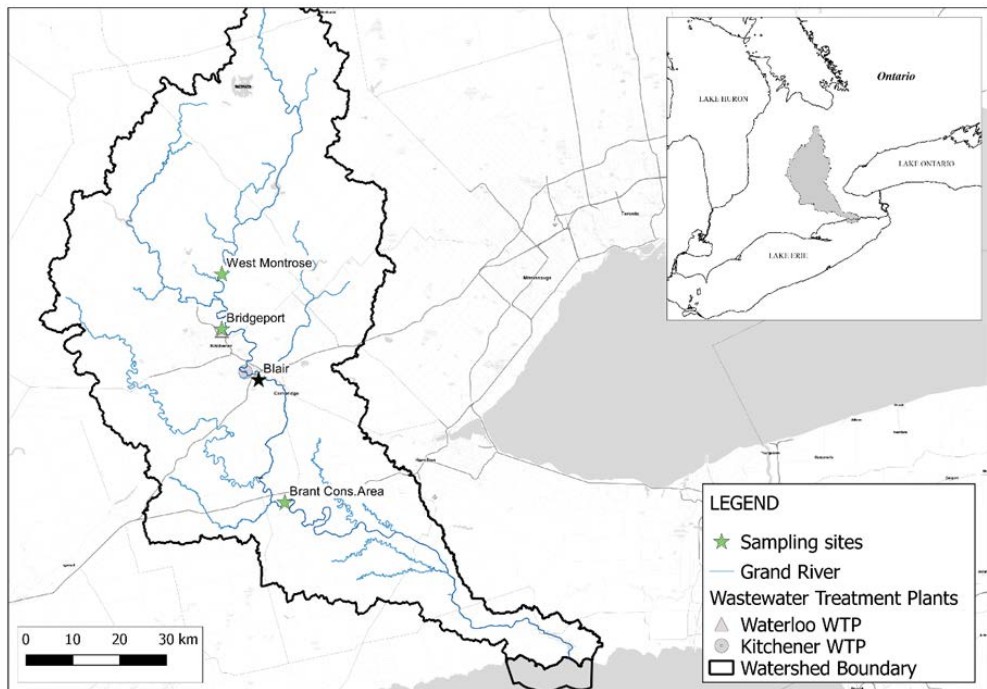

**Figure 1.** Study sites in the central Grand River. Sampling locations West Montrose and Bridgeport represent upstream of the most populated urban area; Blair located within the urban area, 5.7 km downstream of the Kitchener Wastewater Treatment Plant effluent discharge. Brant Conservation Area represents the southern section of the case study.

### 2.2. Sample Collection and Analysis

The number of sampling events per year varied; therefore, average values refer to time-weighted averages, calculated from the total of samples by season per year before and after WTP upgrades. Temperature (°C), pH and electrical conductivity (mS/cm) were measured in situ with multi-parameter probes (Hach HQ40d and YSI 560) previously calibrated. Water samples for concentrations ($NH_4^+$, $NO_2^-$, $NO_3^-$ and TN) were collected in HDPE bottles, stored in ice (cooler) and the filtered in the laboratory (0.45 μm nitrocellulose filter membrane) before analyses, stored in a cold room (4 °C). Total ammonium nitrogen (TAN = $NH_3$ + $NH_4^+$) and nitrite ($NO_2^-$) were measured by colorimetric methods [33], using a UV-VIS Beckman spectrophotometer and a Smartchem 200 Autoanalyzer (±5% precision). When nitrite was below detection limit (0.01 mg N- $NO_2^-$) it was counted as half the detection limit for all calculations. Nitrate ($NO_3^-$) was analysed with an ion chromatograph (Dionex Corp., Sunnyvale, CA, USA, ±5% precision). Total nitrogen (TN) was analyzed by acid combustion in Apollo 9000 Combustion TOC/TN Analyzer (Teledyne Tekmar) and Shimadzu TOC-L Total Organic Carbon Analyzer with TNM-L Total Nitrogen Measuring Unit (precision ±0.3 mg C/N-DOC/TN L-1). Dissolved inorganic nitrogen (DIN) was obtained by adding $NO_2^-$ + $NO_3^-$ + $NH_4^+$. Statistical analyses and graphics were produced with SPSS 13.0 (SAS Institute, Cary, NC, USA).

Data collected from 2010 to 2012 represents before and during upgrade conditions and 2013 represents the year after upgrades conditions. WTP discharge and chemistry were obtained from the Region of Waterloo and from the Water and wastewater monitoring report [31]. River discharge was obtained from the Historical Hydrometric Data Search (Water Survey of Canada) and from the Grand River Information Network (GRIN Open Data Licence v1.0). Meteorological information was obtained from the archives of the University of Waterloo weather station (43.473778 N, 80.557639 W; 334.4 m.a.s.l.; http://weather.uwaterloo.ca/ (accessed on 8 May 2015).

### 2.3. Seasonal Demarcation

The hydrology of the Grand River does not follow the meteorological definition of seasons due to the dominance of the snowmelt period in early spring. Therefore, we proposed seasonal demarcation based on discharge data (from Water Survey of Canada). We selected the location Blair (BL) representing the central Grand River within the urban area and plotted daily discharge ($m^3$/s; Figure 2) from 2010 to 2013 and monthly average discharge (2006–2013) in order to identify season by flow regime. We named low and high flow regime according to deviation of the monthly average from values above or below the annual average discharge (31.7 $m^3$/s; Figure 2). Mid flow refers to discharge slightly above yearly average (around 40 $m^3$/s). Using this hydrograph, we are also able to distinguish wet or dry years by comparing the trends between daily and monthly average discharge. In this manner, seasons were identified in Julian days as shown in Table 1. Additional criteria supporting the cut-off dates are included in Appendix A.

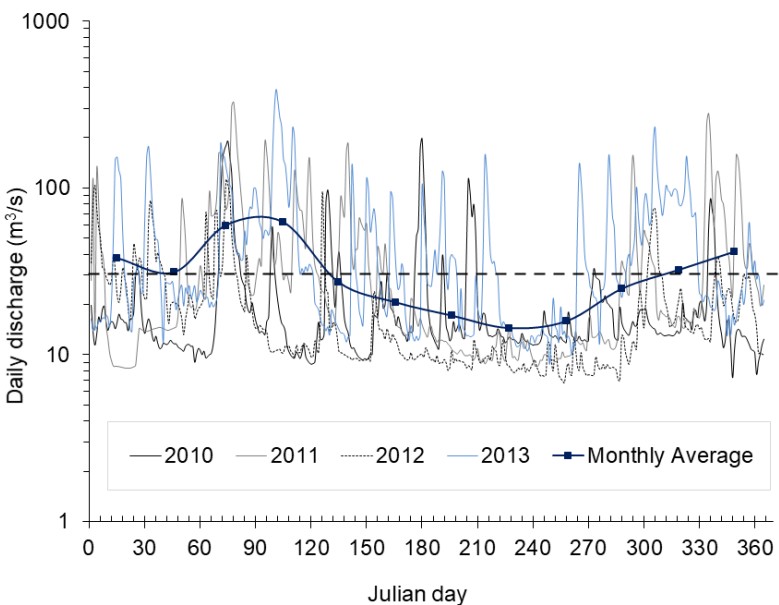

**Figure 2.** Hydrograph of the central Grand River at Blair (146 km from headwaters). Daily discharge (2010 to 2013) compared to monthly average discharge (2006–2013) for seasonal demarcation by flow regime. River discharge was obtained from the Historical Hydrometric Data Search (Water Survey of Canada) and from the Grand River Information Network (GRIN Open Data Licence v1.0).

**Table 1.** Seasonal demarcation (in Julian days) based on monthly average discharge (solid line in Figure 2). Flow regime based on deviations of monthly average discharge relative to yearly average discharge (dotted line in Figure 2). Data from the Grand River Information Network (GRIN Open Data Licence v1.0).

| Season | Start Day | End Day | Duration (Days) | Flow Regime |
|---|---|---|---|---|
| Winter | *320* | *45* | *91* | Average flow |
| Spring–Spring melt | *46* | *134* | *89* | High flow |
| Summer–Fall | *135* | *319* | *185* | Low flow |

For our purposes, above-average annual river discharge is the measurable expression of the thickness of the snowpack before snowmelt (regulating high soil saturation), elevated average annual precipitation (increasing river flow due to surface runoff and driving high groundwater discharge) and increased water release from dams. Due to the volume discharged by the KWTP and the magnitude of the upgrades, a large part of the results here discussed refers to before and after upgrades at the KWTP. However, it is important

to mention that the assumed urban DIN came from the two largest WTPs (Waterloo and Kitchener) in addition to other WTP's located upstream of the Region of Waterloo.

## 3. Results and Discussion

### 3.1. The Impact of the Upgrades at the Kitchener Wastewater Treatment Plant

The overriding change in the operation of the KWTP was reduced ammonium in the effluent, from ~25 mg $N-NH_4^+/L$ to ~5 mg $N-NH_4^+/L$ due to submerged aeration in the oxidation tanks (completed by January 2013). Ammonium concentration consistently below 5 mg $N-NH_4^+/L$ has been observed in the effluent since May 2013. The effects of upgrades are seen in the DO and the DIN concentrations at Blair (5700 m downstream of the KWTP). According to the Grand River Conservation Authority [34], the chemical oxygen demand (COD) of the effluent was reduced from ~125 mg COD/L before upgrades, to 5.8 mg COD/L after upgrades. Similar improvements in water quality after wastewater treatment plant upgrades have been achieved in a temperate river (North Carolina), where 81% reduction in ammonium and 28% reduction in nitrate were achieved, with the resulting increase in dissolved oxygen [35]. The nitrogen load released from the KWTP (Region of Waterloo, *unpublished data*) was lower before upgrades (1681 $\pm$ 216 kg N-DIN/d,) than after upgrades (1905 $\pm$ 236 kg N-DIN/d, Student's t = 2.66, *p* = 0.01,); approximately 220 kg N per day more after completion of the upgrades. This change in nitrogen load might attributed to reduced ammonia volatilization, the increase in volume treated (larger discharge) and the recirculation of the *centrate*, a low-volume, high-concentration liquid result of biosolids dewatering. Water quality is particularly important during periods of low flow due to low oxygen saturation (at Blair) and drinking water withdrawn from the Grand River (at Brantford). Approximately 165,000 $m^3$ per day (5500 kg N/d) are discharged into the Central Grand River (Region of Waterloo, personal communication). The KWTP represents approximately 42% (70,000 $m^3$/d) of that treated sewage discharged into the Central Grand River [34].

### 3.2. Intra-Annual Variability of DIN in the Central Grand River

The general trend observed within a year in the Central Grand River was of elevated DIN concentration during low temperature seasons (namely, winter) and high flow periods (spring and spring melt). Figure 3 shows the behavior of nitrate and ammonium. Ammonium concentrations (TAN) upstream of the urban area were variable but low throughout the year (0.11 $\pm$ 0.25 mg N/L annual average concentration from 2010 to 2013) with some specific increases in late summer and fall, assumed the result of manure application. The locations upstream of the urban area (West Montrose and Bridgeport, representing inputs from agriculture activities) had high concentrations ($\geq$5 mg N/L) in the late fall and winter, whereas concentrations between 2 and 3 mg N/L were observed during spring and summer. The sampling location within the urban area (Blair) showed similar trends; however, this last location had extended periods with nitrate close or above 4 mg N/L, due to its proximity to the KWTP (5700 m downstream of the effluent). Nitrate above 5 mg N/L during winter was also observed at Brantford, 40 km downstream of the KWTP.

Blair, the location close to the KWTP had the highest annual average TAN concentrations in the Central River before upgrades (0.44 $\pm$ 0.39 mg $N-NH_4^+/L$; F = 54.6, *p* < 0.0001, df = 426; Figure 4). Ammonium was particularly high in this sampling location at night during the low flow period in summer nights (0.2 mg N/L, *n* = 9, average concentration in July of 2010, 2011 and 2012) arguably due to low dissolved oxygen and lack of photosynthetic oxygen evolution. Agricultural and urban land uses and the entrance of the tributary Speed River influence the south end of the Central Grand River (Brantford, 187.9 km from headwaters). The TAN annual average concentration was 0.12 mg $N-NH_4^+/L$ from 2010 to 2012, which suggests that, even before upgrades, large part of the ammonium had been assimilated, volatilized or nitrified.

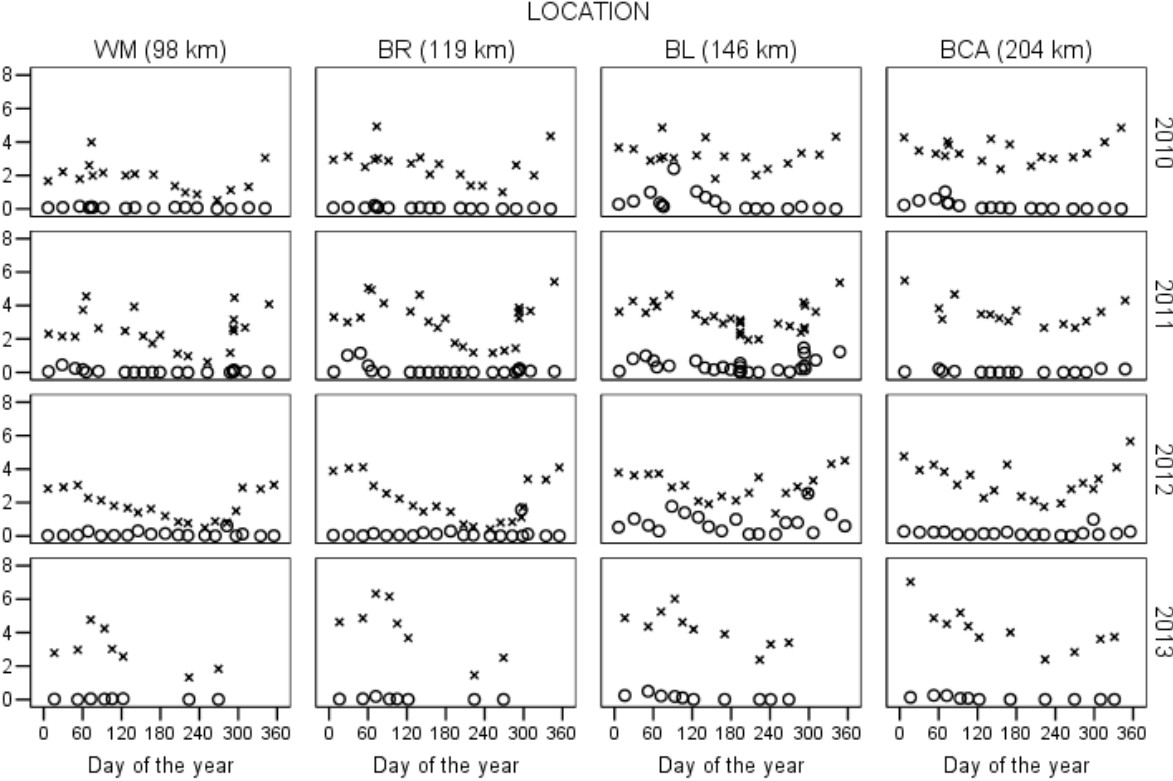

**Figure 3.** Concentration of ammonium ($\bigcirc$, mg $N-NH_4^+/L$) and nitrate ($\times$, mg $N-NO_3^-/L$) quantified in the central Grand River during the period 2010–2013. Sampling locations WM (West Montrose) and BR (Bridgeport) represent upstream of the urban area; BL (Blair) located within the urban area, downstream of the Kitchener Wastewater Treatment Plant effluent discharge; BCA (Brantford) represents the southern section of the study. The year 2013 represents the effects of in-stream N concentration after upgrades to the Kitchener Wastewater Treatment Plant.

Nitrite ($NO_2^-$) was observed to be particularly high year-round at BL (annual average $0.2 \pm 0.05$ mg N/L before upgrades) but was frequently below the detection limit after the upgrades and most of the study period in the rest of the locations. Nitrate ($NO_3^-$) was commonly higher in winter than the rest of the year, and showed an increasing trend as the river flowed downstream. Upstream of the urban area, nitrate ranged from 2.5 to 3.5 mg N/L. Downstream of the KWTP effluent, the $NO_3^-$ annual average concentration at Blair varied from 3.2 to 4.0 mg N/L; the highest annual average concentration was measured after upgrades (Tukey–Kramer HSD q = 2.34, *p* = 0.12; Figure 4). Further downstream, at Brantford, the $NO_3^-$ annual average (4.09 mg N/L) was not different before and after upgrades. Complete dataset available as Table S1 (DIN concentration in the Grand River (ON, Canada), before, during and after WTP upgrades).

Seasonal differences observed in nitrate in the Grand River can be caused by flow because several solutes have a positive concentration–discharge (c-Q) relation, which implies that erosion or runoff is bringing most solutes [36]. Given that variable -Q patterns within one stream has been observed in long term studies [37], some intra-annual differences in DIN concentrations in the Central Grand River may be explained by the variable water temperature among (microbial activity increases with temperature) seasons and the nutrient demand from in-stream plants/algae and actively growing crops within the watershed. The difference in annual average water temperature is likely not as important as the 20 °C difference observed within a year. The seasonal effect of water temperature was observed as nitrate concentrations equal or greater than 4 mg $N-NO_3^-/L$ in most locations of the Central Grand, and ammonium concentrations equal or greater than 0.5 mg $N-NH_4^+/L$ as far as 164 km from headwaters from fall until mid-spring (over winter), concurrent with water temperatures lower than 15 °C (Figure 5). Assimilation for the majority of the mesophilic biota is optimal at 10 °C; thus, reductions in temperature resulted in reduced

assimilation of nitrate in several algal and bacterial of different physiological types [38]. The optimum temperature for nitrification is 15 °C [39]; thus, high nitrate could be expected in low water temperature conditions. It has been found intense denitrification (and maximum N$_2$O concentration) during summer with high water temperature and low dissolved oxygen conditions [25]. A laboratory study assessed denitrification rates in riverbed sediments which are not nitrate limited, where they found that lowering the temperature to 4 °C resulted in an approximately 77% decrease in N$_2$O production rates [40]. Assuming that summer hypoxia will no longer be common due to the upgrades in the WTPs, it is possible that preventing hypoxic conditions might reduce denitrification in the Grand River, potentially leading to higher-than-expected nitrate concentrations, especially during warmer periods.

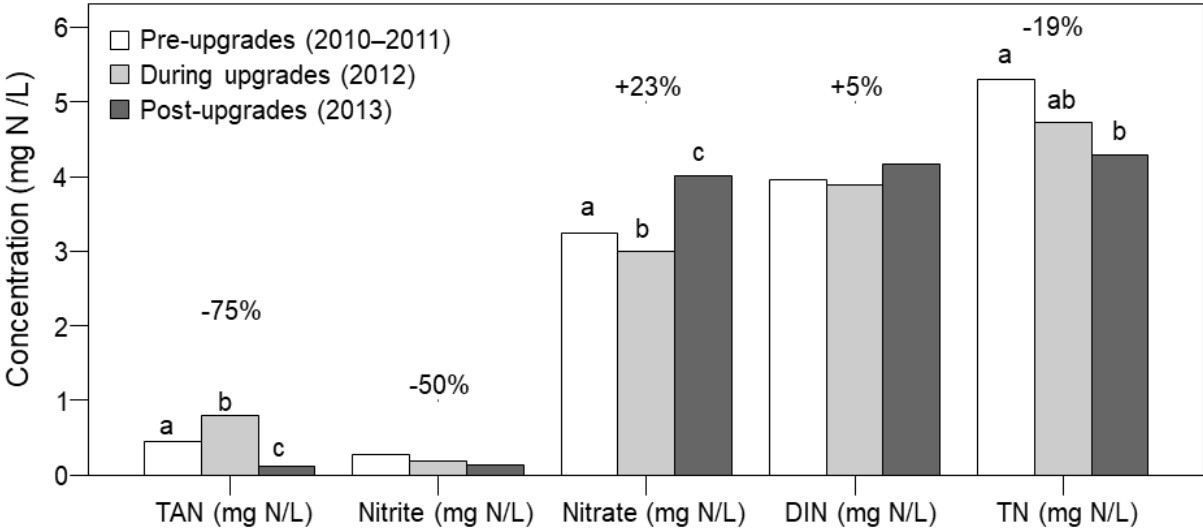

**Figure 4.** Changes in annual average concentration of water quality parameters at the location Blair, 5.7 km downstream of Kitchener wastewater treatment plant. Values with different letters (a, b, c) are statistically different: post hoc test LSD (a = 0.05). Pre-upgrades (2010–2012, *n* = 66) and post-upgrades (2013, *n* = 17). Samples represent daytime concentrations. River discharge obtained from the Grand River Information Network (GRIN Open Data License v1.0).

The fact that nitrate inputs from agricultural catchments during the non-growing season occur simultaneously with low temperature in the Grand River likely results in high nitrate (and ammonium) concentrations in the Grand River during winter through spring (Figure 5). Ammonium and nitrate concentrations usually decrease when biological assimilation by crops is more intense (during the growing season), but increase because of active tile drainage when crops are not growing anymore. Additionally, during the high-water table seasons (late fall, winter and spring), reduced nitrate assimilation, high runoff and active tile drainage resulted in nitrate being mobilized from the agricultural sub-catchments into the tributaries of the Grand River; thus, leading to large agricultural nitrate contributions. Groundwater upstream of Brantford before upgrades, had nitrate concentrations between 0.05 to 5.0 mgN-NO$_3^-$/L (median = 3.8 mgN-NO$_3^-$/L [41]); thus, additional nitrate from groundwater discharge could also play a role in seasonal variability if groundwater discharge is lowest in summer.

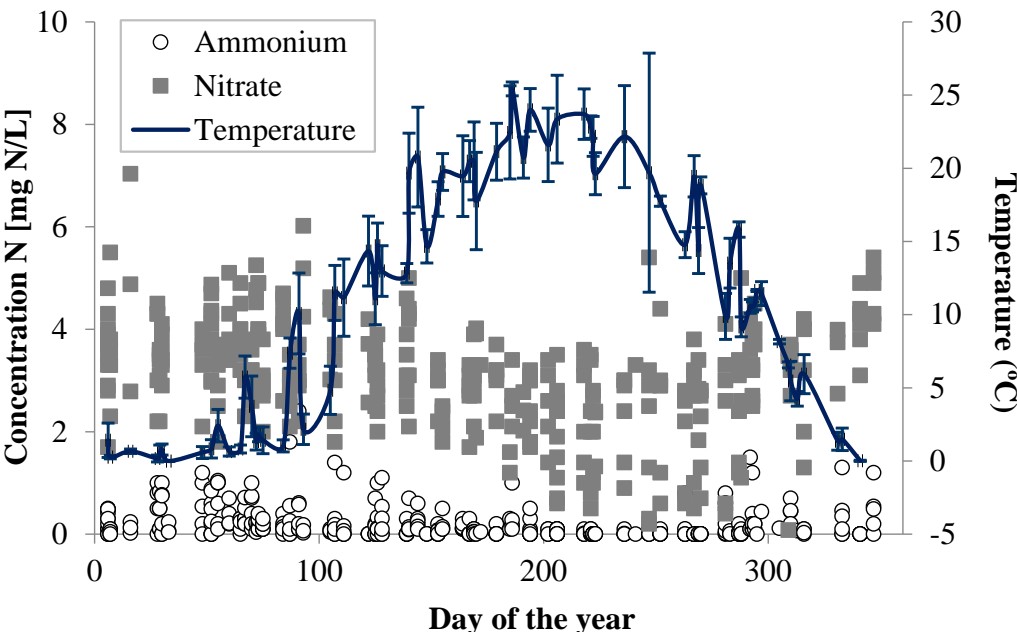

**Figure 5.** Ammonium and Nitrate concentration (mg N/L) and river temperature in the Central Grand River from 2010 to 2013. Data represents location from 98 km from headwaters (West Montrose) to 204 km (Brantford).

*3.3. Inter-Annual Variability of DIN in the Central Grand River*

The DIN concentrations observed in the central Grand River downstream of the KWTP effluent discharge is considered to be driven by the KWTP effluent: river dilution ratio, which changes due to the variable volumes in river discharge among wet and dry years. During years with flow below historical average (i.e., dry years) the relative contribution of the KWTP effluent was approximately 8% of the Central Grand River discharge downstream of the KWTP effluent (at Blair, 146 km from headwaters). During years with flow above historical normal (wet years), the effluent of the KWTP represented between 3 and 5% of the Grand River discharge. During the year 2012, several samples collected at Blair during the night had ammonium concentrations above 1 mg N/L and extremely high values of 2 mgN-$NH_4^+$/L in late summer 2012. On the other hand, wet years (2011 and 2013) had above-historical average river discharge, which enhanced dilution of the N inputs from WTP's.

Nitrate concentrations were expected to decrease to 3 mg N/L at Brantford (47 km downstream of the KWTP effluent, 204 km from headwaters) due to dilution, biological uptake and denitrification [42]. However, in the fall of 2013, nitrate concentration in the Grand River downstream of the KWTP was between 3.3 and 4 mg N/L, surpassing the target value. We noted an overall reduction in $NO_3^-$ downstream of the KWTP; however, it was not always below the $NO_3^-$ target value. The magnitude of nitrate increase based a one-dimensional, dynamic nutrient and dissolved oxygen water quality model (Grand River Simulation Model) was predicted to be around 1.1 mg N/L higher than upstream locations. The modeled scenarios for summer low flow consider simultaneous increases in cumulative upstream sources (i.e., increase in population served by WTP's) and a 10 to 25% reduction in non-point sources [42]. However, with the samples collected between 2010 and 2013, the nitrate increase used for modelling purposes (1.1 mg N/L) is likely to be surpassed in the summer during dry years. In the event of extreme low river flow, long-term exposure to high nitrate concentration is likely to represent important impacts on sensitive aquatic organism [43] in addition to issues arising from exceeding N permissible limit in drinking water.

Since the DIN had an increasing trend as the Grand River flows southwards (Figure 6), it is likely that the Grand River is not assimilating the entire DIN generated in the agri-

cultural and urban sub-catchments and DIN is farther downstream [44]. Complementary to the inorganic nitrogen species, the dissolved organic nitrogen (DON) was an important component of the total dissolved nitrogen measured in the Grand River, possibly of agricultural provenance. The dissolved organic nitrogen (DON) was not measured as extensively as DIN in this study, its concentration downstream of the urban area was not significantly different before and after upgrades (t = 2.03, *p* = 0.08). This DON accounted for an annual average of 24% (±12%) of the TN measured in the central Grand River. These measurements are in good agreement with previous reports in urban-agricultural landscapes [45,46]. DON contribution from treated effluent varies largely [47]; however, DON from WTP's has been reported to be highly bioavailable [48]. DON is actively taken up by biota in nitrogen-poor environments [49]; given that the DIN is abundant, probably it is not in high demand in the Central Grand River.

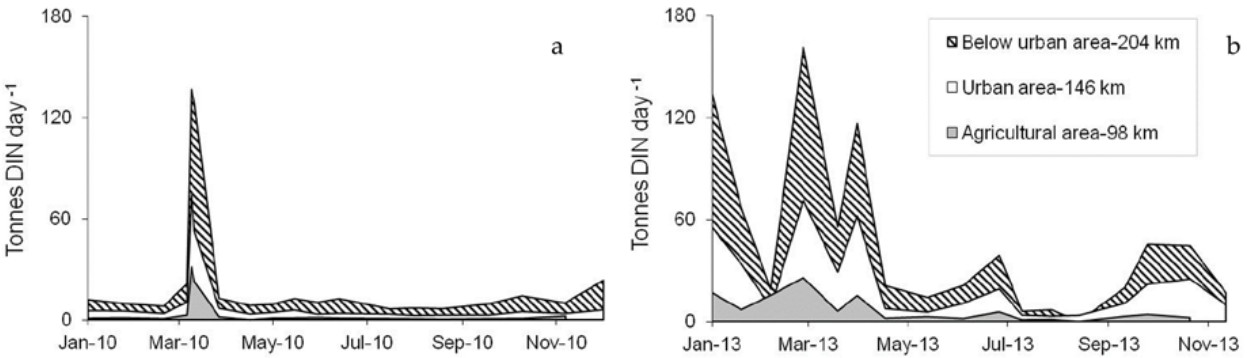

**Figure 6.** Dissolved inorganic nitrogen loads (tonnes DIN per day) in three sections of the Central Grand River in 2010 (**a**) and 2013 (**b**). Agricultural area-West Montrose (02GA034); Urban Area–Blair (02GA048); Below urban area–Brantford (02GB001). Discharge data obtained from Water Survey Canada and GRCA. Distance in kilometers from headwaters.

### 3.4. Loads and Concentrations—A Different Answer to a Different Question

DIN loading clearly responds to variable river discharge (Figure 6). The addition of agricultural and urban inputs leads to the peaks observed at the urban area (146 km from headwaters). A warm year with close-to-average base flow (2010), showed a single, clear peak during spring melt. On the other hand, a very wet year with above-average base flow (2013) had large variability in DIN loads throughout the year. The increase in N observed downstream of the urban area (204 km) is assumed the cumulative effect of all agricultural and urban inputs, in addition to tributaries and groundwater discharges. High river discharge can also represent reduced contact time with the riverbed; thus, leading to reduced N assimilation and promoting that reactive N lingered in the water column for longer periods. Nitrogen loading is expected to increase in the Central Grand River as the population served by the WTP's increases, or if additional agricultural nitrogen is being added to the watercourses. The Grand River Watershed Water Management Plan [22] estimated nitrate load upstream of the urban area as follows: agricultural creeks (50%), the Conestogo River (40%), the Shand dam (8%) and septic systems (2%). Tributaries upstream of the urban area might contribute to as much as 60 kg N per day during spring melt high flow (March–April) [50].

Monitoring nitrogen concentration is the quintessential measurement of water quality as it relates to the permissible limits of these particular substances. However, comparing concentrations without considering the river discharge is challenging and could be misleading due to the differential dilution of the nutrients and solutes transported across the watershed under different discharge regimes and among years, due to the fact that concentration is a parameter largely influenced by weather conditions. Comparing concentrations among years is necessary to satisfy the regulatory framework and guidelines set by environmental authorities; compliance with such guidelines ensures the proper functioning of the river as an ecosystem and as the recipient and conveyor of treated effluent from

urban areas. On the other hand, loads are particularly important when producing nutrient balances at watershed scales and are relevant for downstream receiving water bodies to address environmental effects, best management practices, geochemical budgets and impacts of climate change [51–53]. Wet years (above-average annual discharge) would likely have high N loads at critical dates (such as spring melt and high precipitation events), whereas dry years (below-average annual discharge) might have high nitrate concentrations downstream of point sources, especially below the urban area due to low base-flow. Changes in the river discharge entails changes in the fluxes of elements, not necessarily because of erosion, but also resulting from human activities [54,55]. Accurate loading calculations require frequent water quality monitoring and discharge data (stage or flow velocity are also useful if the channel morphology is known); therefore, both monitoring strategies strength the capacity of doing better predictions and nutrient modeling [56,57]. Our results provide valuable and useful information that would allow regulatory agencies and water managers, to evaluate the effectiveness and the impacts of the upgrades completed on WTPs depending on the purposes and objectives of the diverse final users.

## 4. Conclusions

The upgrades completed in the KWTP succeeded in reducing ammonium concentration in the effluent. However, the nitrate concentration in the Central Grand River downstream of the KWTP effluent after upgrades in the fall of 2013 was above the Grand River nitrate target value of 3 mg $N-NO_3^-/L$). The before-after approach used in this study allowed us to understand the dynamics of the DIN due to changes in the operation of a wastewater treatment plant in an anthropogenically impacted river. Intra-annual variations include seasonal effects such as changes in river discharge and water temperature. High ammonium and nitrate concentrations concurrent with water temperatures lower than 15°C were observed fall until mid-spring (over winter). Intra- and inter-annual variations are driven by the river discharge; increases in river discharge caused dilution of nutrients. Dry years would likely have low dilution rate of the effluent in the river and likely an increase in nitrate concentrations to higher-than-expected levels. Differences in the DIN concentrations between seasons and between years were not only attributed to changes in the quality of the WTP's effluent, but also a result of upstream nitrate inputs from agricultural sources. The limited amount of data after the upgrades limits the interpretation; yet, monitoring, including years with above historical average and below historical average base flow, would be desirable to capture the complete inter-annual variability and put in context the impact of the WTP's upgrades. High flow–high load events are of special interests when evaluating nutrients export and producing nutrient balances at watershed scales; thus, understanding changes in the nutrient status of urbanized rivers will support the design and implementation of effective monitoring strategies in areas with similar geographic and climatic conditions to those observed in southwestern Ontario.

**Supplementary Materials:** The following are available online at https://www.mdpi.com/article/10.3390/nitrogen2020010/s1, Table S1: DIN concentration in the Grand River (ON, Canada), before, during and after WTP upgrades.

**Author Contributions:** E.C. Conception and design, acquisition, analysis and interpretation of data, involved in preparing the manuscript. M.S.R. Acquisition, analysis and interpretation of data, involved in preparing the manuscript. R.J.E. Acquisition, analysis and interpretation of data. S.L.S. Conception and design, analysis and interpretation of data, involved in preparing the manuscript, funding acquisition. All authors have read and agreed to the published version of the manuscript.

**Funding:** This research was funded by CONACyT scholarship 191667 (E. Cejudo). The NSERC provided funding for the study design and data analysis through the strategic project (STPGP 381058-2009) 'Linking cycles of O2, P and N in impacted rivers: ecosystem response to changes in wastewater treatment plant effluent'. Manuscript preparation completed at the University of Waterloo and Centro de Investigación Científica de Yucatan A.C. (Project Catedras CONACYT 2944).

**Institutional Review Board Statement:** Not applicable.

**Informed Consent Statement:** Not applicable.

**Data Availability Statement:** The data presented in this study are openly available in Mendeley data at http://dx.doi.org/10.17632/ydbggrjtt9.1 (accessed on 10 April 2021).

**Acknowledgments:** Thanks to P. Aukes, F. Cummings, D. Dilworth, J. Harbin, R. Hutchins, S. Sine, E. Westberg J. Zheng and staff from the Environmental Geochemistry Laboratory (University of Waterloo) who were involved in data collection. R.O. Aravena for his support and critical comments. The Grand River Conservation Authority provided access to gauge station and discharge data available under Grand River Conservation Authority's Open Data Licence v1.0.

**Conflicts of Interest:** The authors declare no conflict of interest. The funders had no role in the design of the study; in the collection, analyses, or interpretation of data; in the writing of the manuscript, or in the decision to publish the results.

## Appendix A

**Table A1.** Relevant meteorological information from 2010 to 2013. Data obtained from the University of Waterloo weather station (43.4738 N, 80.5576 W). Discharge ($m^3/s$) from Water Survey of Canada, stations West Montrose (WM-02GA034), Doon (BL-02GA048) and Brantford (BCA-02GB001). Long-term average discharge WM = 16 $m^3/s$ (45 y), BL = 31 $m^3/s$ (8 y) and BCA = 59 $m^3/s$ (65 y).

| Year | Comments | Temp °C | | Historical avg °C | Year Avg °C | Historical Precip. avg [mm] | Total Year Precip. [mm] | Year Snow avg [cm] | Total Year Snow [cm] | Mean Q ($m^3$/s) Water Survey of Canada | | |
| | | High | Low | | | | | | | WM | BL | BCA |
|---|---|---|---|---|---|---|---|---|---|---|---|---|
| 2010 | Dry year. 5th warmest year in history, 1.4 °C above average. Warmer spring. One week in July and one in August were notably above average. 11 days over 30 °C. Abundant rain: maximum one day rain: 65 mm. 1 in 10 years precipitation. 46.1 °C highest humidex (July 7th) | 33.1 | −21.5 | *11.89* | 13.11 | *904* | 879.3 | *159.5* | 77.5 | 11.5 | 22 | 45.3 |
| 2011 | Wet year, 4th wettest year since 1914. Wettest April ever. The first half of the year was 1 °C colder than average; the second half was 2.5 °C warmer than average. | 35.7 | −28.8 | *11.89* | 12.6 | *904* | 1146.4 | *159.5* | 165.5 | 17.4 | 38 | 81.6 |
| 2012 | Dry, consistently warm year. Hot March, over 7.5 °C warmer than average. The second warmest year since 1914. July 2011 to June 2012, the warmest 12 month period in the history. A large part of the year was drier than average. 56 mm one-day precipitation event (June 1st). 46.9 °C humidex (July 21st) | 33.5 | −18.3 | *11.89* | 14.31 | *904* | 782.7 | *159.5* | 86.5 | 10.7 | 19.2 | 44 |

**Table A1.** *Cont.*

| Year | Comments | Temp °C | | Historical avg °C | Year Avg °C | Historical Precip. avg [mm] | Total Year Precip. [mm] | Year Snow avg [cm] | Total Year Snow [cm] | Mean Q (m³/s) Water Survey of Canada | | |
|---|---|---|---|---|---|---|---|---|---|---|---|---|
| | | High | Low | | | | | | | WM | BL | BCA |
| **2013** | Wet year. Year with more precipitation in 99 years record. 94.1 mm in one-day precipitation period (September). February, second snowiest record in the region. Late cold winter. 186 days without frost, longest frost-free season since 1915. 47.6 °C highest humidex (July 17th) | 34.7 | −24 | *11.89* | 11.92 | *904* | 1204.7 | *159.5* | 179 | *n.a.* | *n.a.* | *n.a.* |

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
