# Peer review of "Intra- and Inter-Annual Variability in the Dissolved Inorganic Nitrogen in an Urbanized River before and after Wastewater Treatment Plant Upgrades: Case Study in the Grand River (Southwestern Ontario)"

_nitrogen, doi:10.3390/nitrogen2020010_

Round 1

Reviewer 1 Report

This study investigate the change of inorganic nitrogen compounds in the river before and after WWTP upgrade. It is very interesting study. However, it is difficult to accept with present state.

  1. More in-depth discussion should be required.
  2. unit missed in Table 1
  3. please provide basic information of WWTP process. 

Author Response

Response to Reviewer 1 Comments

  1. More in-depth discussion should be required.

Response 1: We appreciate he suggestion. We have revisited some sections of the discussion that could be improved after a second read and some additions have been made. We hope that these changes seem fit for Reviewer 1 since there were not specific sections mentioned.

  1. unit missed in Table 1

Response 2: We have updated the Table caption as we notice the lack of cross-reference to figure 2. The units for the table al Julian days, they are indicated in the caption (L130-132). We appreciate the observation, we have added the units “days” to the Duration column header. The column “Flow regime”, is a qualitative description of the deviation from the monthly average discharge relative to yearly average discharge, as such, it does not have units (L136).

  1. please provide basic information of WWTP process. 

Response 3: We have added additional information about both wastewater treatment plants in lines 71 and 75.

Reviewer 2 Report

General comments:

The manuscript deals with "monitoring inorganic nitrogen in the Grand river before and after running the modified wastewater treatment plant". The manuscript has provided a sufficient detail about the effects of a WWTP on the amount of nitrogen dissolved in a river.

Inorganic nitrogen pollution (such as ammonia) has been remarked a serious environmental problem because it may cause eutrophication, a toxic problem to aquatic environments. 

1. Page 1, Line 14; "..that assessed changes in N.." Before using the abbreviation, you need to mention to the full name. Therefore, you may mention it in the first line as "External nitrogen (N) inputs originating from.."

2. Page 1, Line 15; "..wastewater treatment plant (WTP).." Usually, researchers have used the WWTP as an abbreviation for the wastewater treatment plant".

3. Line 13 "...we present...", Line 15 "We measured..", Line 22 "We suggest..", Line 24 "we propose..."!! Using several pronouns in a scientific paper is not sounding good!

4.  Page 1, Line 17; "These upgrades reduced the ammonium concentration.." Mention to the values.

5. Page 1, Line "...being higher during the low temperature.." Mention to the values.

6. Write keywords alphabetically.

7. Page 4, Line 78; "..sampling was completed from 2010 to 2013.." Long time ago!! The data used was out of date!! I think the data needs to be updated!!

8. The quality of figure 3 should be improved.

Author Response

Response to Reviewer 1 Comments

  1. Page 1, Line 14; "..that assessed changes in N.." Before using the abbreviation, you need to mention to the full name. Therefore, you may mention it in the first line as "External nitrogen (N) inputs originating from..

Response 1: Thank you kindly for the observation and the suggestion, we have added the letter N as nitrogen is first mentioned in L 11.

  1. Page 1, Line 15; "..wastewater treatment plant (WTP).." Usually, researchers have used the WWTP as an abbreviation for the wastewater treatment plant".

Response 2: Thanks for the suggestion, we agree that the conventional abbreviation for wastewater treatment plant is WWTP; however, given that, we profusely use the acronym with the initial of the city (Kitchener or Waterloo), we prefer to avoid the awkwardness of WWWTP, similar to KWWTP, also to keep the abbreviation simple. We kindly ask the reviewer and the Assistant Editor to allow us to keep the current format as in L15, L63, L71, L74.

  1. Line 13 "...we present...", Line 15 "We measured..", Line 22 "We suggest..", Line 24 "we propose..."!! Using several pronouns in a scientific paper is not sounding good!

Response 3: Thanks much for the observation. In order to make the abstract readable, pronouns were eliminated or replaced, so the sentences read “This research presents (L13), Changes in the DIN effluent discharge, in-river concentrations and loads were observed (L15), This suggest (L22)”. L25 kept the wording “we propose” as it expresses the authors’ opinion about what regulatory agencies and water managers might do.

  1. Page 1, Line 17; "These upgrades reduced the ammonium concentration.." Mention to the values.

Response 4: We have added the values in L 18 “concentration in the river from 0.44 to 0.11 mg N-NH4+/L (year average)”

  1. Page 1, Line "...being higher during the low temperature.." Mention to the values.

Response 5: thanks for the suggestion; the sentence now reads, “being higher during the low temperature (> 10° C) and high flow seasons (spring and spring melt)” L20.

  1. Write keywords alphabetically.

Response 6: The keywords are now in alphabetic order (L28 Dissolved Inorganic Nitrogen; effluent; seasonality; urban; wastewater). Thanks for the observation.

  1. Page 4, Line 78; "..sampling was completed from 2010 to 2013.." Long time ago!! The data used was out of date!! I think the data needs to be updated!!

Response 7:  Right, we understand the concern. The data reported in this paper was part of a doctoral thesis, as such, only the data in that document is included in this contribution. The funding for the Grand River project looking into coupled N, P and O2 cycles finished (NSERC strategic project STPGP 381058-2009), and so the sampling and analyses of samples. There were sampling efforts in some specific sites and dates afterwards, but the new data (after 2013) is being used somehow differently by someone else. In addition, it would be very important to publish the data from 2010-2013 so any new or recent data can be compared on an historical perspective by both researchers and water managers. We hope that Reviewer 2 understands the situation.

  1. The quality of figure 3 should be improved.

Response 8: thanks for the suggestion; we have improved the quality of Figure 3. (L199)

Round 2

Reviewer 1 Report

author well responded to all comments, 

but please thoroughly check typo and error 

Reviewer 2 Report

Reviewers' comments have been addressed.